# Selecting Suitable Tree Species for Direct Seeding to Restore Forest Ecosystems in Northern Thailand

Khuanphirom Naruangsri [1], Wasu Pathom-aree [2], Stephen Elliott [1,*] and Pimonrat Tiansawat [1,*]

1   Forest Restoration Research Unit, Department of Biology, Faculty of Science, Chiang Mai University, 239 Huaykaew Road, Mueang District, Chiang Mai 50200, Thailand; khuanphirom_n@cmu.ac.th

2   Department of Biology, Faculty of Science, Chiang Mai University, 239 Huaykaew Road, Mueang District, Chiang Mai 50200, Thailand; wasu.p@cmu.ac.th

*   Correspondence: stephen_elliott1@yahoo.com (S.E.); pimonrat.t@cmu.ac.th (P.T.)

**Abstract:** To upscale restoration of tropical forest ecosystems, direct seeding—sowing seeds directly into the ground—is potentially a more cost-effective technique than tree planting. However, its success is limited by seed predation, the harshness of environmental conditions on restoration sites and particularly by a lack of information about suitable tree species. Therefore, this study tested the suitability of 23 native forest tree species for direct seeding, to restore a biodiversity-rich, upland, evergreen forest in northern Thailand. Three replicate seed batches of each species were sown randomly in two degraded sites and in a tree nursery under controlled conditions. Seed removal and germination were monitored weekly until no further germination had occurred for more than a month. Subsequently, seedling yield, growth and species performance score were also monitored at appropriate intervals. Nine months after sowing, seed removal differed significantly among species but was generally low, with a cross-species average of 3.4% ($\pm$0.5 SE). Seed size was negatively correlated with seed removal. Eight species failed to germinate. Seed germination percentage varied widely among species. Cross-species average germination of the 15 species that germinated was 25% ($\pm$6.2 SE). Two species (*Adenanthera microsperma* and *Alangium kurzii*) were ranked as having high germination (>50%), five species (*Choerospondias axillaris*, *Spondias pinnata*, *Diospyros glandulosa*, *Melia azedarach* and *Phyllanthus emblica*) had medium germination (20–50%) and eight species had low germination (<20%). Following the first dry season, two of the fifteen germinated species failed to establish. Germination and establishment were influenced by seed size, seed storage behavior and successional status. *A. microsperma*, *S. pinnata* and *C. axillaris* are recommended for direct seeding based on their high species performance index values. This study further concluded that selecting desiccation-tolerant seeds, particularly those with medium-to-large sizes, could increase the chances of successful seedling establishment.

**Keywords:** direct seeding; forest restoration; native tree species; species performance

## 1. Introduction

At the UN Climate Change Conference COP26 in 2021, leaders of 141 countries, covering 91% of the world's forests, committed their nations to "conserve forest ecosystems and accelerate their restoration" and to co-operate in "halting and reversing forest loss and land degradation" [1,2]. This ambitious declaration, and the global initiatives that preceded it, e.g., the Bonn Challenge [3], the One Trillion Tree Initiative platform [4] and the UN Decade on Ecosystem Restoration (www.decadeonrestoration.org/), etc., have spurred forest-restoration projects around the world on unprecedented scales [5]. In the humid tropics, where conditions for plant growth are ideal, simply protecting and enhancing natural forest regeneration can promote forest recovery, where such regeneration is dense. However, naturally regenerating sites often become dominated by a handful of pioneer tree species because of a lack of nearby tree-seed sources and a loss of seed-dispersing

animals (since up to 80% of tropical tree species may be animal-dispersed) [6]. Under such circumstances, recovery of tree species richness and biodiversity is limited [7–9]. Consequently, tree planting remains the primary technique employed to meet the lofty ambitions of global restoration initiatives.

Planting a wide variety of native forest tree species is recommended, to rapidly accumulate biomass and recover forest structure, biodiversity and ecological functioning in restoration forests [10]. However, tree planting is costly and labor-intensive. It entails collecting seeds from the reference forest ecosystem, establishing a nursery to produce containerized saplings (usually 30–50 cm tall in plastic bags) and transporting them to restoration sites. Tree planting and subsequent weeding and fertilizer application are all highly labor-intensive. Furthermore, sites available for restoration are mostly on steep, difficult terrain, far from vehicular access, i.e., those unsuitable for agriculture.

Direct seeding circumvents some of these logistical limitations and provides a means to upscale forest restoration projects, to meet the needs of the global initiatives mentioned above [7,11]. The method involves simply sowing or burying tree seeds directly into the ground. People become the primary seed-dispersal agents, where natural seed dispersal is limited. Direct seeding requires no nursery costs, and it is far less labor-intensive than conventional tree planting; transportation costs are also much reduced [7,12,13]. It is easier to carry bags of seeds onto steep or remote sites than to haul baskets of containerized saplings. Moreover, seedlings from direct seeding often grow better in the field than nursery-produced saplings because they develop better root systems in situ and transplantation shock is avoided [14].

Direct seeding has been widely trialed in several countries with mixed results [15]. For example, Silva et al. [16] reported an average emergence of around 52% for mixed species of tree seeds sown into neotropical savannas, whilst Grossnickle and Ivetić [11] reported 17% establishment, following direct seeding of tropical forest tree species. In Thailand, the potential of direct seeding for forest restoration was tested in northern seasonally dry forests (e.g., Woods and Elliott [12], Tunjai [17], Hossain et al. [18], Naruangsri et al. [14], Waiboonya [19]) and in southern evergreen forests (e.g., Tunjai [17]), with average seedling establishment ranging from 0 up to 89%. Success appears to be highly species-specific. In southern Thailand, Tunjai [17] concluded that large round seeds (>5 g) with thick seed coats (>0.4 mm) are likely to be successful in the seasonally dry tropics. Waiboonya [19] reported that the optimal time to sow seeds for restoration of upland evergreen forest in northern Thailand was at the beginning of the rainy season.

In a meta-analysis of 30 studies, including both tropical and temperate forests (but none in Thailand), Ceccon et al. [20] reported that overall seed germination was 20%, and approximately 28% of the studied species exceeded 20% seedling establishment. Outcomes were not significantly affected by climate, species successional status nor the application of pre-sowing treatments. Success increased with seed size and with the application of physical protection from seed predators. More recently, in a global bibliometric analysis of 81 publications on direct seeding for forest restoration, Souza [21] reported that forests, established by direct seeding, are rarely monitored for long-term outcomes. He concluded that the technique has great potential to attain restoration goals but that it is insufficiently studied and is, therefore, a promising area for research, to determine its applicability around the world. He attributed its lack of wide adoption in the tropics thus far [11,20] to low seedling emergence, establishment and growth, low seed availability and a lack of knowledge of seed biology (desiccation tolerance, orthodoxy vs. recalcitrance etc.), suitable storage conditions, optimal seeding densities and times—all limitations that may ultimately be minimized or overcome by appropriate species choice. For restoring tropical forests, species selection for direct seeding is more complex and challenging than it is for tree planting. Susceptibility to seed predation is crucial, along with germinability, tolerance of very young seedlings to the harsh conditions on exposed deforested sites and their resilience following damage [10,22].

Therefore, we explored the direct-seeding suitability of 23 tree species, ranging widely in seed size, for restoring an upland evergreen forest in northern Thailand. The observations covered the period from seed sowing to seedling establishment, in order to compare the effects of seed predation, seed germination, seedling yield and seedling growth on direct seeding success. The aim was to increase understanding of how species selection for direct seeding might be optimized.

## 2. Materials and Methods

### 2.1. Study Sites

Field experiments were conducted at two sites under the authority of Nong Hoi Royal Project in Mae Rim District in Chiang Mai, northern Thailand (Figure 1). At each site, the field trials covered 7800 m².

The first was a degraded site near Mon Cham viewpoint (MC) at 1300 m altitude (18°56′18.0″ N, 98°49′16.7″ E) and mostly surrounded by remaining evergreen forest (Figure S1c,d). Since 2012, the site had been reserved for forest restoration as part of a national flood-prevention program. It had previously been used for intensive cultivation of strawberries with heavy use of pesticides and had subsequently become densely covered with herbaceous weeds, particularly exotic grasses (e.g., *Brachiaria* sp.), bracken fern (*Pteridium aquilinum*) and members of the Asteraceae (*Chromolaena odorata*, *Ageratina adenophora*, etc.) partially shaded beneath sparse tree cover.

The second site was near Ban Mae Khi (BMK), an agroforestry research plot at 925 m altitude (18°57′34.0″ N, 98°48′33.4″ E), where some bamboos and fruit trees had been planted, with a central ridge, largely bare of tree cover (Figure S1a,b). The site supported a diverse ground flora, and was dominated by grasses (*Imperata cylindrica*, *Chrysopogon aciculatus*, etc.), *Chromolaena odorata* and *Urena lobata*.

The original vegetation at both sites had been evergreen forest, cleared 40–60 years previously. The total rainfall during the experimental period (2019) was 1638.5 mm (averaging 125.2 mm per month), with a 21 °C average annual temperature and an 87% air humidity. During the experimental period (July 2019 to May 2020), there was a three-month period from January to March 2020 without rainfall [23] (Figure S2).

Seed germination tests were also conducted at a research nursery located in Doi Suthep-Pui National Park, Chiang Mai, Thailand (18°48′3.7″ N, 98°54′59.6″ E, at about 1000 m altitude) under controlled conditions for comparison.

### 2.2. Studied Species

Twenty-three native tree species (from 19 families), representative of northern Thailand's upland evergreen forest (600–1500 m above sea level) or EGF [24], were selected for the experiments. These included 14 species with orthodox seeds, 8 species with recalcitrant seeds and one species of unknown seed-storage behavior. Even though the forest type was EGF, 13 of the individual species studied were actually deciduous. Propagule dry mass (seeds or pyrenes) varied among the species from 0.02 g to 4.30 g. All the species had been proven suitable for EGF restoration by conventional tree planting [25], and all fruited and produced viable seeds about the start of the rainy season, which is the optimum period for direct seeding [17].

More than 1000 seeds of each species were collected from at least five maternal trees, all of which were indigenous to Doi Suthep-Pui National Park, Chiang Mai, Thailand. Mature, fleshy fruits and dry pods were collected directly from tree branches, with some being picked up from the ground. The seeds were then cleaned, air-dried and stored at 4 °C until they were used [19]. The storage time before sowing depended on the collection date, which varied across the species. Moreover, 10–20 seeds per replicate were inspected by standard cut tests, to estimate the percentage of seeds with healthy-looking endosperm and embryos as a proxy for viability (Table 1).

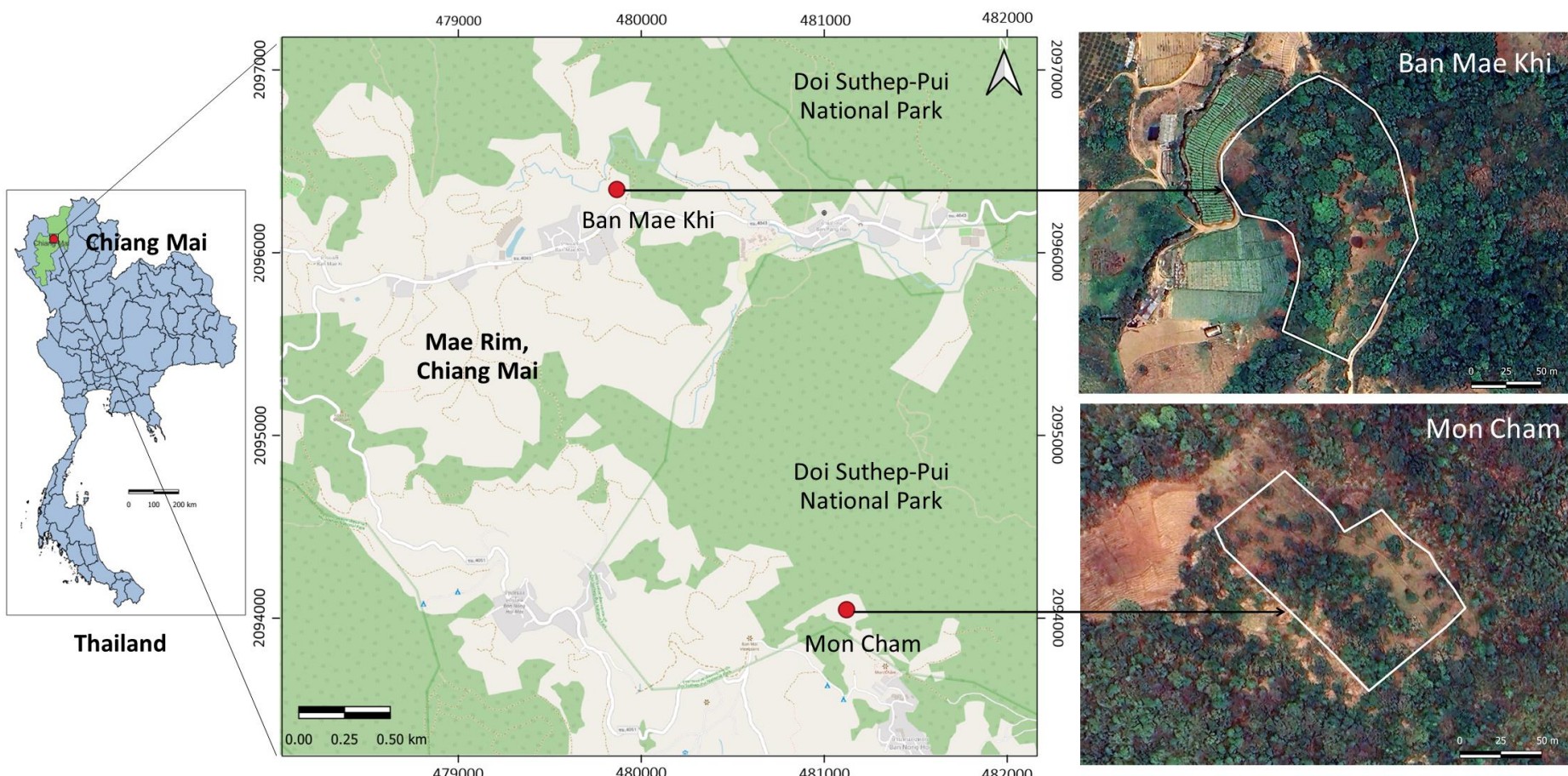

**Figure 1. Left**: the location of the study sites: Chiang Mai, Thailand; **center**: (red dot) the two field sites: Ban Mae Khi and Mon Cham; **right**: plot boundaries and landscape context.

**Table 1.** Characteristics of the 23 native tree species used for the direct seeding experiment, arranged by increasing seed mass.

| No. | Scientific Name | Family | Propagule Mass [1] (g) ± SD | | Storage Behavior [2] | Leaf Habits [3] | Successional Guilds [4] | Collection Date | % Healthy Seed [5] |
|---|---|---|---|---|---|---|---|---|---|
| 1 | *Phyllanthus emblica* L. | Euphorbiaceae | 0.017 ± 0.005 | Small | Orthodox | Deciduous | P | Mar 2019 | 95 |
| 2 | *Hovenia dulcis* Thunb. | Rhamnaceae | 0.028 ± 0.005 | Small | Orthodox | Deciduous | P [4] | Oct 2018 | 90 |
| 3 | *Acrocarpus fraxinifolius* Wight & Arn. | Leguminosae | 0.034 ± 0.005 | Small | Orthodox | Deciduous | P | Jun 2019 | 100 |
| 4 | *Magnolia baillonii* Pierre | Magnoliaceae | 0.047 ± 0.014 | Small | Orthodox | Deciduous | C | Sep 2019 | 100 |
| 5 | *Melia azedarach* L. | Meliaceae | 0.050 ± 0.007 | Small | Orthodox | Deciduous | P [4] | Mar 2019 | 100 |
| 6 | *Balakata baccata* (Roxb.) Esser | Euphorbiaceae | 0.055 ± 0.005 | Small | Recalcitrant | Evergreen | P | Jul 2019 | 100 |
| 7 | *Phoebe cathia* (D.Don) Kosterm. | Lauraceae | 0.083 ± 0.020 | Small | Recalcitrant | Evergreen | C | Jul 2019 | 60 |
| 8 | *Adenanthera microsperma* Teijsm. & Binn. | Leguminosae | 0.104 ± 0.019 | Medium | Orthodox | Deciduous | C | Mar 2019 | 100 |
| 9 | *Artocarpus lacucha* Wall. ex Roxb. | Moraceae | 0.173 ± 0.039 | Medium | Recalcitrant | Deciduous | C | Jun 2019 | 85 |
| 10 | *Alangium kurzii* Craib | Alangiaceae | 0.175 ± 0.018 | Medium | Orthodox | Evergreen | C | Jul 2019 | 100 |
| 11 | *Prunus cerasoides* D.Don | Rosaceae | 0.183 ± 0.039 | Medium | Orthodox | Deciduous | IP [4] | May 2019 | 100 |
| 12 | *Diospyros glandulosa* Lace | Ebenaceae | 0.257 ± 0.062 | Medium | Orthodox | Evergreen | IP [4] | Dec 2018 | 100 |
| 13 | *Cassia bakeriana* Craib | Leguminosae | 0.269 ± 0.039 | Medium | Orthodox | Deciduous | P | May 2019 | 100 |
| 14 | *Syzygium fruticosum* DC. | Myrtaceae | 0.375 ± 0.071 | Medium | Recalcitrant | Evergreen | C | Jul 2019 | 90 |
| 15 | *Gmelina arborea* Roxb. ex Sm. | Verbenaceae | 0.519 ± 0.098 | Medium | Orthodox | Deciduous | P | Apr 2019 | 90 |
| 16 | *Sarcosperma arboreum* Buch.-Ham. ex C.B.Clarke | Sapotaceae | 1.342 ± 0.210 | Medium | Recalcitrant | Evergreen | IC [4] | Jun 2019 | 100 |
| 17 | *Choerospondias axillaris* (Roxb.) B.L.Burtt & A.W.Hill | Anacardiaceae | 1.434 ± 0.275 | Medium | Orthodox | Deciduous | I | Aug 2019 | 100 |
| 18 | *Polyalthia viridis* Craib | Annonaceae | 1.553 ± 0.218 | Medium | Recalcitrant | Evergreen | C | Jun 2019 | 80 |
| 19 | *Garcinia cowa* Roxb. ex Choisy | Guttiferae | 1.755 ± 0.364 | Medium | Recalcitrant | Evergreen | C | Jun 2019 | 100 |
| 20 | *Quercus brandisiana* Kurz | Fagaceae | 1.776 ± 0.536 | Medium | Recalcitrant | Evergreen | C | Jun 2019 | 100 |
| 21 | *Sapindus rarak* DC. | Sapindaceae | 1.946 ± 0.253 | Medium | Orthodox | Deciduous | IP [4] | Apr 2019 | 100 |
| 22 | *Scleropyrum pentandrum* (Dennst.) Mabb. | Santalaceae | 2.525 ± 0.672 | Large | - | Evergreen | - | Jun 2019 | 100 |
| 23 | *Spondias pinnata* (L.f.) Kurz | Anacardiaceae | 4.263 ± 0.678 | Large | Orthodox | Deciduous | P | Dec 2018 | 100 |

[1] Seed size (dry mass): small (0.01–0.099 g); medium (0.1–2 g); large (more than 2.0 g), from Waiboonya [19]. [2] Forest Restoration Research Unit [26], Waiboonya [19] and the Seed Information Database [27]. [3] Gardner et al. [24] and FORRU-CMU's database. P = pioneer, IP = intermediate pioneer, I = intermediate, IC = intermediate climax, C = climax, from Manohan et al. [28]. Other species from [4] Gardner et al. [24] and FORRU-CMU's database. [5] Percent healthy seed calculated from proportion of full-embryo seeds per total sample from cut test.

Seed morphological traits: width, length, depth, coat thickness and wet and dry mass were measured for all studied species. The seeds were classified based on their storage behavior using FORRU [26], Waiboonya [19] and the Seed Information Database [27]. Furthermore, the successional guild and leafing habit of each species were assigned, based on Manohan et al. [28], Gardner et al. [24] and FORRU-CMU's database. The species were categorized by seed size using the size classes of Waiboonya [19] (Table 1).

### 2.3. Experimental Design and Data Collection

Three replicates, each of 20 seeds, were hand-sown into both field sites during the rainy season of 2019. Bamboo tubes (about 10 cm long and 5–10 cm diameter) were buried 5 cm deep into the soil near bamboo marker sticks established a meter apart from one another. In each tube, one seed was pressed into the soil and buried about 0.5 cm deep. A tag, indicating the identity of the seed in each tube, was attached. A total of 1380 seeds from 23 species were sown in each study site.

Percent seed removal and germinated seeds were recorded weekly for nine months from sowing on 29 July 2019. This study used seed removal to indicate the intensity of seed predation [29]. Seed removal comprised both destroyed and dispersed seeds, both of which reduced the number of seeds remaining in the study plots. Germination was defined as emergence of a primary root, cotyledon or hypocotyl visible on the surface of the soil. Monitoring ceased when no further germination had occurred for more than a month.

Three replicates of twenty seeds of each species were also sown in a tree nursery in modular germination trays under 50%–70% shade in parallel to the field experiments. This determined germination rates under ideals conditions and without predation. Germination was recorded in the same way as in the field trials.

During the first year of the field experiments, weeds were removed by hand and fertilizer was applied in November 2019 and again in May 2020 (at the end and beginning of the rainy season, respectively). The number of surviving seedlings was recorded during such maintenance procedures. Root collar diameter (RCD)—stem diameter where shoot meets root—was measured at the widest point, using Vernier-scale calipers. Seedling (or sapling) height (from root collar to apical meristem) and crown width (at broadest axis) were measured with a ruler (as outlined in the FORRU protocol [25]).

### 2.4. Data Analysis

All statistical analyses were performed using R version 4.0.2 [30], applying a significance level of $p < 0.05$.

#### 2.4.1. Seed Removal and Germination and Seedling Yield

Seed removal, germination and seedling yield were calculated as a percentage of the total number of seeds sown. A generalized linear model (GLM) with a logit link function was applied to determine the significance of treatment effects on seed removal, germination and seedling survival (yield). The independent variables were species and sites. The dependent variable was the proportion of seed removal, germination and survival. When significant effects were found, significant differences between means were determined by Tukey's HSD test.

#### 2.4.2. Seedling Growth

Seedling height, root collar diameter (RCD) and crown width (CW) were monitored twice on 24 November 2019 and after the dry season on 22 May 2020 (over a total of 180 days). For each species, relative growth rates (RGRs) were determined for all seedlings, using differences in height (RGR-H), root collar diameter (RGR-RCD) and crown width (RGR-CW) between the two monitoring dates and the formula below:

$$\text{RGR (\% per year )} = \frac{ln(\text{final size}) - ln(\text{initial size})}{\text{number of days between measurements}} \times 365 \times 100$$

Daily proportional growth, relative to the average plant size over the measurement interval, was multiplied by 365 (to derive an annual value) and by 100 to convert to a percentage, as modified from Hoffmann and Poorter [31]. Analysis of variance (ANOVA) was used to test whether the means of seedling performance variables (absolute values and RGR-H, RGR-RCD and RGR-CW) differed between the study sites and among the species. When ANOVA indicated differences, Tukey's HSD test was used to determine which of the means were significantly different from one another.

### 2.4.3. Relative Performance Index (RPI)

To determine whether species were suitable for direct seeding, a relative performance index (RPI) was devised, which combined both seedling yield and growth into a single indicator. Seedling yield was the proportion of seeds that became established seedlings. The average RCD (mm) of each species was used to represent seedling size, as it is closely and positively correlated with seedling height, crown width and plant biomass [31]. A raw performance index was calculated by multiplying the relative seedling yield by relative size, using RCD. The score was transformed into a relative score (RPI) by expressing each raw score as a percentage of the highest species score. RPI is unitless.

### 2.4.4. Effect of Species Traits on Direct Seeding

Pearson's correlation analysis and principal component analysis (PCA) were used to explore relationships between various species traits and field data (continuous data) (Figures S4–S6). The most influential variables, thus identified, were then used to construct a generalized linear model (GLM) to determine the most predictive traits and their relative effects on seed removal, germination and survival. The independent variables were dry propagule mass (continuous data), seed/propagule storage behavior (orthodox vs. recalcitrant) and successional guild (pioneer vs. climax). The dependent variables were seed removal, germination and survival. Seed dry mass was used to represent seed physical characteristics because it was strongly correlated with seed dimensions and seed-coat thickness (Figures S4 and S6). In addition, for each species trait, ANOVA was used to detect differences in mean seedling growth, including crown width (CW), height (H), root collar diameter (RCD) and relative growth rate (RGR) among the groups. When ANOVA indicated significant differences, Tukey's HSD tests were used to determine which means were significantly different from each other. Furthermore, the non-parametric Kruskal–Wallis test was applied, to test the effects of seed size and successional guild on RPI.

## 3. Results

### 3.1. Seed Removal

Percent seed removal varied among the 23 species and two sites. Zero removal was recorded for five species: *A. microsperma*, *Q. brandisiana*, *S. arboreum*, *S. pentandrum* and *S. pinnata*. For 18 species, mean percent removal ranged from 0.8% (±1.2 SE) (for *A. fraxinifolius*, *C. axillaris*, *D. glandulosa* and *P. cathia*) up to a maximum of 9.2% (±5.8 SE) for *C. bakeriana* (Figure 2). Percent removal, averaged across the species, was <5% at all study sites. The highest removal was recorded at BMK (4.3%, ±0.5 SE), followed by MC (2.5%, ±0.4 SE) and the tree nursery (0.1%, ±0.1 SE). The GLM indicated a significant effect of study site on percent seed removal but no species effect (coefficient estimate ± SD = −7.0 ± 1.0, $z = -7.0$, $p < 0.001$). The probability of seed removal at the three study sites ranged from 0 to 0.04.

Pearson's correlation analysis and PCA indicated that seed dry mass was negatively correlated with seed removal (Figures S4 and S6). Linear regression also indicated that percent seed removal significantly decreased with increasing seed mass (coefficient estimate ± SE = −0.6 ± 0.3, t = −2.2, $p = 0.04$), but the relationship was extremely weak (R-squared = 0.2). Smaller seeds (e.g., *C. bakeriana* and *H. dulcis*) were more likely to be removed than larger ones (e.g., *S. pentandrum* and *S. pinnata*).

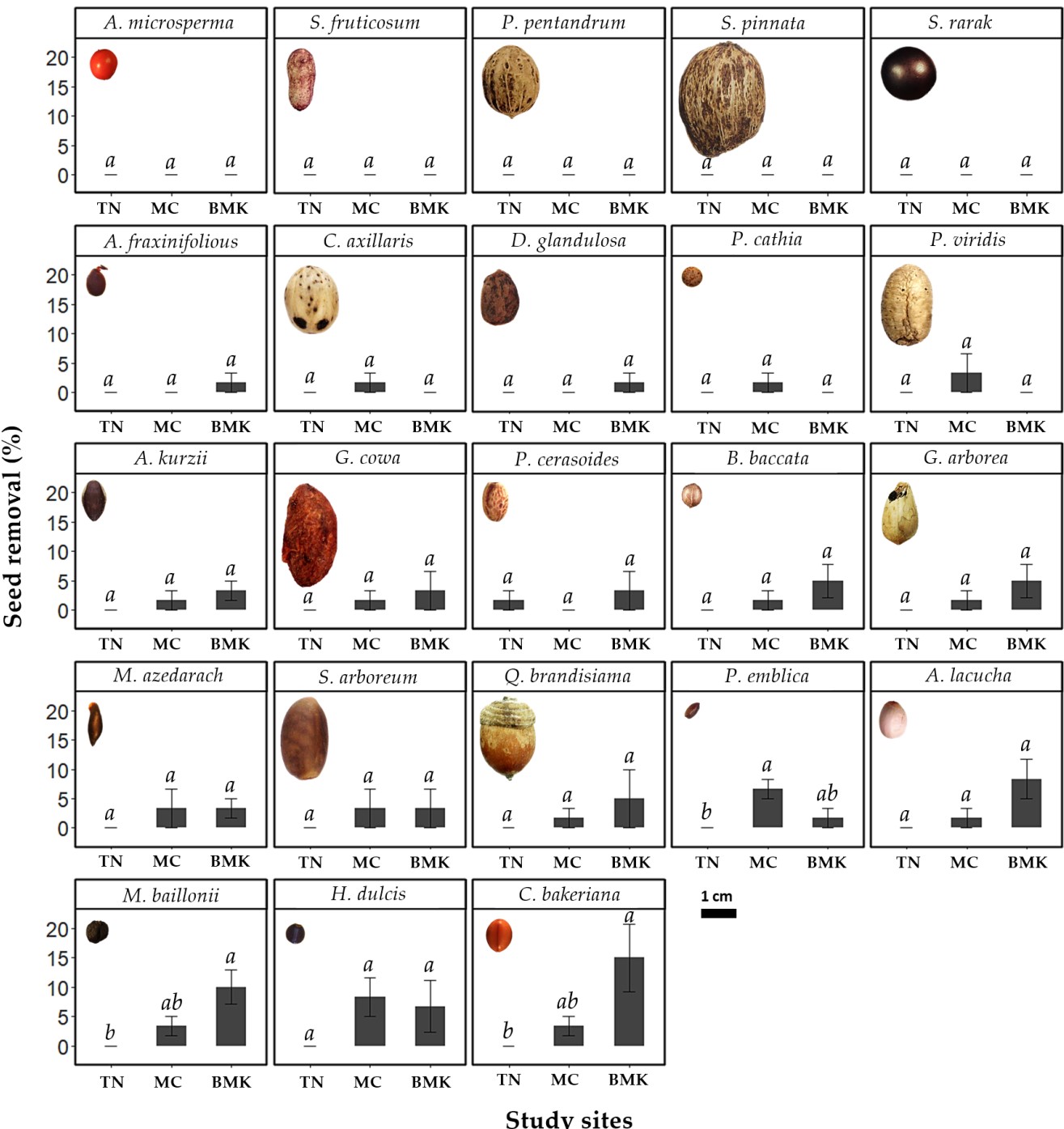

**Figure 2.** Seed removal (% ±1 SE) compared among sites: tree nursery (TN), Ban Mae Khi (BMC) and Mon Cham (MC). Five species in the top row of the figure had no seed removal. Species panels are arranged in order of increasing removal rates. Columns not sharing the same letter *a* and *b* indicate significant differences among sites.

*3.2. Seed Germination*

Eight species (35% of the studied species) failed to germinate. Non-germinating species were excluded from further analyses. The species were categorized into three groups: high, medium and low germination (Figure 3). Eight species had low percent seed germination (<20%, with a group average of 8.8% (±1.4 SE)), ranging from 0.6% (±0.6 SE) for *B. baccata* to 16.1% (±8.7 SE) for *P. cathia*. Five species achieved moderate seed

germination (20–50% germination, with a group average of 25.4% (±1.7 SE)), ranging from 21.7% (±5.9 SE) for *P. emblica* to 30.6% (±13.8 SE) for *C. axillaris*. Only two species attained germination percentages higher than 50%: *A. kurzii* at 68.8% (±7.5 SE), with *A. microsperma* being the highest at 85% (±3.5 SE) (group average: 76.4% (±7.6 SE)). The GLM showed a significant interaction effect between species and study site on seed germination (coefficient estimate ± SE = −3.4 ± 0.7, $z_i$ = −4.6, $p$ < 0.001). Seed germination was differed among sites, indicating a site-specific effect. *C. axillaris*, *S. pinnata* and *P. emblica* achieved higher seed germination in the field sites, whereas the seeds of *A. kurzii* and *H. dulcis* germinated better in the tree nursery (Figure 3).

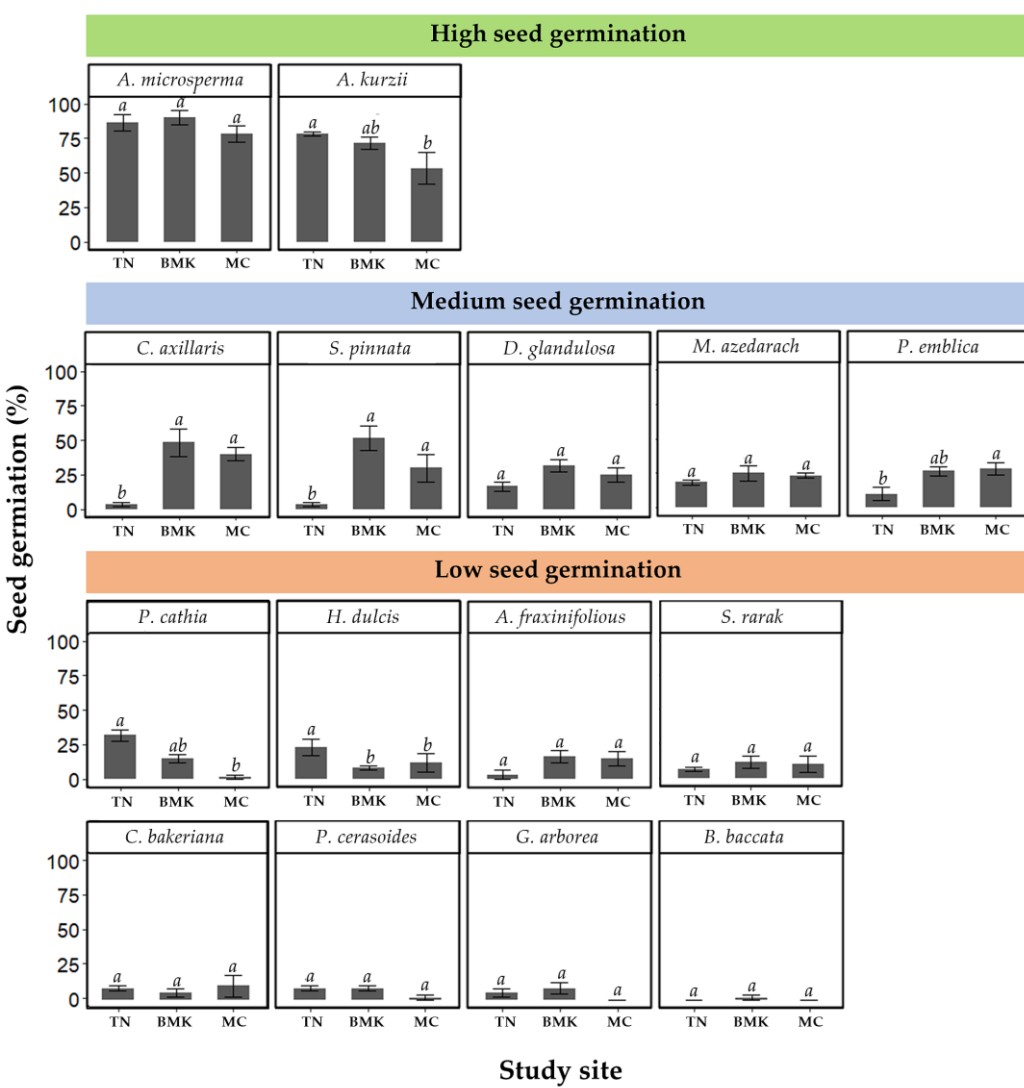

**Figure 3.** Percentage seed germination (±1 SE) across study sites. Eight species, which failed to germinate, are not included. Columns not sharing the same letter *a* and *b* indicate significant differences among sites.

Among all the species studied, the GLM showed a significant effect of successional guild on seed germination (coefficient estimate ± SE = −5.0 ± 1.4, $z_i$ = −3.6, $p$ = 0.003) and a significant interaction effect of successional guild and seed size (coefficient estimate ± SE = 17.0 ± 6.6, $z_i$ = 2.6, $p$ = 0.02). Late successional species had a higher germination probability. Furthermore, percent germination increased with increasing seed size. Moreover, the GLM also indicated a significant interaction effect of seed size and seed storage behavior on seed germination (coefficient estimate ± SE = −58.7 ± 27.1, $z_i$ = −2.2, $p$ = 0.049).

The effect of seed storage behavior on the probability of seed germination was marginally significant (coefficient estimate $\pm$ SE = 10.2 $\pm$ 4.7, $z_i$ = 2.2, $p$ = 0.05). Germination failure (zero probability of germination) was more likely for recalcitrant seeds than for orthodox ones. The probability of seed germination decreased with decreasing seed size.

### 3.3. Seedling Yield

The overall seedling yields across the surviving species were similar between the two field sites, averaging 21.2% ($\pm$14.0 SE) at MC and 20.1% ($\pm$20.3 SE) at BMK. The GLM indicated no significant effect of study site on seedling yield (coefficient estimate $\pm$ SE = $-0.0 \pm 0.1$, $z_i$ = $-0.1$, $p$ = 0.95). However, the effect of species on seedling yield was statistically significant ($p < 0.05$).

Of the 15 species that germinated, two (*B. baccata* and *P. cathia*) failed to establish any seedlings. Differences in seedling yield among the other 13 species were statistically significant ($p < 0.05$), such that the species could be divided into three groups. A single species stood out as having by far the highest seedling yield: *A. microsperma* (66.7%, $\pm$8.3 SE). Four species had moderate seedling yields (with a group average of 25.2% ($\pm$3.5 SE)), ranging from 17.5% ($\pm$0.8 SE) for *P. emblica* to 33.3% ($\pm$8.3 SE) for *S. pinnata*. Nine had poor seedling yields below 15%, ranging from 5% ($\pm$0 SE) for *G. arborea* to 15% ($\pm$0 SE) for *D. glandulosa* (with a group average of 10.3% ($\pm$1.3 SE)) (Table 2).

**Table 2.** Seedling yields, size variables and corresponding relative growth rates (RGR).

| Species | $n$ | %Yield | RCD | | Height | | CW | |
|---|---|---|---|---|---|---|---|---|
| | | Mean $\pm$ SE | Mean $\pm$ SE | %RGR $\pm$ SE | Mean $\pm$ SE | %RGR $\pm$ SE | Mean $\pm$ SE | %RGR $\pm$ SE |
| *A. microsperma* | 72 | 66.67 $\pm$ 8.33 [a] | 2.2 $\pm$ 0.0 [b] | 57 $\pm$ 2.02 [bc] | 12.8 $\pm$ 0.3 [cd] | 83.64 $\pm$ 1.8 [cd] | 13.5 $\pm$ 0.5 [cd] | 74.0 $\pm$ 3.6 [bc] |
| *S. pinnata* | 32 | 33.33 $\pm$ 8.33 [b] | 2.8 $\pm$ 0.0 [b] | 25.19 $\pm$ 6.21 [c] | 11.2 $\pm$ 0.4 [d] | 50.9 $\pm$ 9.7 [de] | 9.6 $\pm$ 0.8 [d] | 44.2 $\pm$ 6.3 [c] |
| *C. axillaris* | 23 | 28.33 $\pm$ 1.67 [b] | 2.4 $\pm$ 0.1 [b] | 136.08 $\pm$ 18.75 [ab] | 22.4 $\pm$ 1.6 [a] | 133.99 $\pm$ 12.3 [b] | 19.0 $\pm$ 0.9 [ab] | 124.1 $\pm$ 5.5 [ab] |
| *A. kurzii* | 20 | 21.67 $\pm$ 1.68 [c] | 1.8 $\pm$ 0.1 [b] | 70.65 $\pm$ 13.64 [bc] | 9.6 $\pm$ 0.7 [d] | 113.52 $\pm$ 4.0 [bc] | 8.0 $\pm$ 0.4 [d] | 75.2 $\pm$ 20.4 [bc] |
| *P. emblica* | 17 | 17.50 $\pm$ 0.83 [cd] | 1.8 $\pm$ 0.1 [b] | 64.09 $\pm$ 7.43 [bc] | 18.4 $\pm$ 1.6 [abc] | 74.75 $\pm$ 12.2 [cde] | 15.4 $\pm$ 1.4 [bcd] | 75.3 $\pm$ 12.1 [bc] |
| *M. azedarach* | 11 | 15.00 $\pm$ 5.00 [de] | 2.6 $\pm$ 0.3 [b] | 47.37 $\pm$ 7.34 [bc] | 20.4 $\pm$ 2.6 [ab] | 101.68 $\pm$ 15.9 [bcd] | 22.6 $\pm$ 3.7 [a] | 146.4 $\pm$ 20.4 [a] |
| *D. glandulosa* | 8 | 15.00 $\pm$ 0.00 [ef] | 2.2 $\pm$ 0.1 [b] | 69.99 $\pm$ 6.35 [bc] | 11.6 $\pm$ 0.4 [d] | 84.56 $\pm$ 11.2 [cd] | 11.2 $\pm$ 1.1 [cd] | 33.0 $\pm$ 12.1 [c] |
| *S. rarak* | 8 | 11.25 $\pm$ 3.75 [ef] | 4.0 $\pm$ 0.4 [a] | 47.08 $\pm$ 20.70 [bc] | 17.0 $\pm$ 1.3 [abcd] | 65.75 $\pm$ 23.5 [cde] | 17.0 $\pm$ 2.0 [abcd] | 26.9 $\pm$ 6.6 [c] |
| *H. dulcis* | 2 | 10.00 $\pm$ 0.00 [ef] | 2.9 $\pm$ 0.5 [ab] | 160.84 $\pm$ 66.88 [a] | 26.2 $\pm$ 9.4 [a] | 236.16 $\pm$ 42.4 [a] | 11.6 $\pm$ 4.6 [cd] | 136.7 $\pm$ 68.4 [ab] |
| *P. cerasoides* | 2 | 10.00 $\pm$ 0.00 [fg] | 2.4 $\pm$ 0.3 [b] | 52.61 $\pm$ 37.65 [bc] | 25.5 $\pm$ 1.5 [a] | 65.96 $\pm$ 7.6 [cde] | 17.5 $\pm$ 0.1 [abc] | 25.5 $\pm$ 13.9 [c] |
| *C. bakeriana* | 8 | 10.00 $\pm$ 2.5 [fg] | 1.9 $\pm$ 0.1 [b] | 15.34 $\pm$ 5.21 [c] | 8.2 $\pm$ 1.1 [d] | 20.37 $\pm$ 7.7 [e] | 7.3 $\pm$ 1.0 [d] | 50.3 $\pm$ 29.2 [bc] |
| *A. fraxinifolius* | 4 | 6.25 $\pm$ 1.25 [g] | 1.8 $\pm$ 0.4 [b] | 157.41 $\pm$ 13.07 [a] | 10.1 $\pm$ 2.0 [d] | 107.58 $\pm$ 14.7 [bcd] | 8.9 $\pm$ 2.6 [d] | 79.8 $\pm$ 2.0 [abc] |
| *G. arborea* | 1 | 5.00 [g] | 2.9 [ab] | 139.2 [ab] | 14.0 [bcd] | 58.3 [cde] | 12.0 [cd] | 37.0 [c] |

$n$ = number of surviving trees (315 in total). Species are ordered by descending seedling yield. [a–g] Values not sharing the same superscripts within columns are significantly different among species.

Based on the GLM, three species traits—seed storage behavior, successional guild and seed size—significantly influenced seedling yield without any interaction effects. The seedling yields significantly increased with increasing seed size (coefficient estimate $\pm$ SE = 0.4 $\pm$ 0.2, $z_i$ = 2.4, $p$ = 0.03). The seedling yield was 35% for large seeds and 6% for small seeds. Furthermore, pioneer species had a lower seedling yield compared with climax species (coefficient estimate $\pm$ SE = 1.9 $\pm$ 0.6, $z_i$ = $-3.4$, $p$ = 0.004). Orthodox species had a significantly higher seedling yield (17%) than recalcitrant species did (zero seedling yield) (coefficient estimate $\pm$ SE = 1.8 $\pm$ 0.8, $z_i$ = 2.2, $p$ = 0.04).

### 3.4. Seedling Growth

Seedling growth varied greatly among the 13 surviving tree species, nine months after seed sowing. ANOVA indicated significant differences in mean seedling height, CW and RCD among species. *H. dulcis* ($n$ = 2) grew the tallest. *M. azedarach* ($n$ = 11) had the broadest canopy. *S. rarak* ($n$ = 8) achieved the highest mean RCD (Table 2).

RGRs of height, CW and RCD exceeded 50% per year for most species. Fast-growing species were *C. axillaris* and *H. dulcis*, with RGRs of RCD, height and CW exceeding 100% per year (Table 2). Furthermore, two other species: *A. fraxinifolius* and *M. azedarach*, also achieved fast growth, with RGR-CWs and RGR-Hs exceeding 100% per year. In contrast,

*S. rarak* and *S. pinnata* were slow-growing, despite having large seedlings at nine months (Table 2).

### 3.5. Relative Performance Indices

 *A. microsperma* attained the highest raw performance score and was assigned as the 100 benchmark. RPIs of the other species that established ranged from 7 to 64. Only one of them attained an RPI that exceeded half that of the highest-performing species: *S. pinata*, with an RPI of 64. The next highest performing species was *C. axillaris,* which scored 46 (Figure 4). Relatively low-performing species, with RPIs lower than 20 were: *P. cerasoides*, *C. bakeriana, G. arborea* and *A. fraxinifolius*. The effects of seed size and successional guild on the relative performance index were not significant.

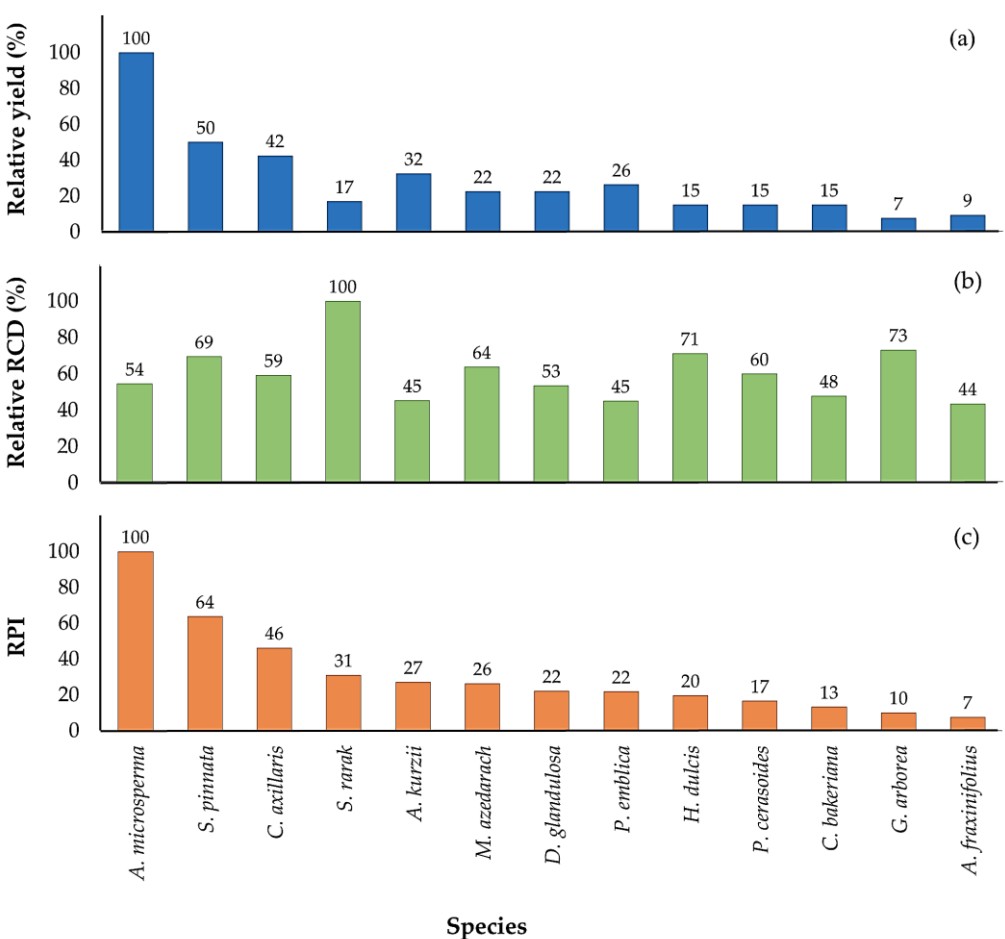

**Figure 4.** Comparison of relative rank score of seedling yield (**a**), relative seedling root collar diameter (RCD) (**b**) and relative percentage performance index (RPI) (**c**) across studied species, ranking from the highest to lowest RPI.

### 4. Discussion

 This study evaluated the suitability of 23 tree species for direct seeding, to restore upland evergreen forest ecosystems in northern Thailand. We investigated the intensity of seed predation and species performance, in terms of seedling yield and growth after the first dry season. Although seed predation was low, 10 out of the 23 species studied failed to establish at all. The seedling yields of those that did establish were mostly low (averaging 20%). This study revealed an interplay of various factors that contributed to low seedling establishment, including effects from study sites, species traits (seed size and storage behavior) and successional guild.

### 4.1. Seed Removal

Small seeds tended to be more vulnerable to seed removal than larger ones, most probably due to seed predation. This agrees with Dylewski et al. [32], who reported that in tropical forests, seed removal rates decrease with increasing seed mass. This may be because smaller seeds are easier to move than large ones, and they tend to lack protective structures, such as thick coverings [33,34]. None of the largest propagules: *S. pinata* and *S. pentandrum*, were removed from the experiment. They are both multiple-seeded pyrenes with tough coverings derived from the fruit endocarp. This observation suggests that large propagules, particularly those with hard coverings, are resistant to predation [34].

However, the results of this study contrasted with those of a predator-exclusion experiment performed at the same Mon Cham plot on 29 July 2015–26 July 2016, during which up to 100% removal was recorded. Large seeds were lost to rodents, but small seeds remained untouched [14]. The fact that the seeds in the present study were more widely spaced and buried deeper than in the previous study might explain the contrasting results. Wide spacing is known to substantially reduce seed predation by rodents [33,35].

### 4.2. Seed Germination

Seed germination probability increased with increasing seed mass. This was consistent with previous research [36–38]. Food reserves within the endosperm of seeds support germination and early seedling growth [39]. Larger seeds typically have a higher concentration of nitrogen and phosphorus than small seeds [40], which can facilitate early seedling development, even where light and nutrients are limiting [41,42]. Small seeds, such as *A. fraxinifolius* and *M. baillonii*, have limited seed resources. Their seedlings therefore perform better, where high sunlight accelerates photosynthesis during seedling emergence and early development, making them less dependent on seed reserves (Table 3). The seeds of such species must therefore be deposited close enough to the soil surface to receive full sunlight for successful emergence [42–44]. The heterogeneity of canopy cover across the study site and the reduced light availability within bamboo tubes may have affected the potential for seed germination, especially of light-demanding species (Table 3).

Seed-storage behavior influenced germination success. The seeds of desiccation-sensitive species (*S. fruticosum*, *S. arborea*, *G. cowa*, *P. cathia*, *P. viridis* and *Q. brandisiana*) failed to germinate, both in the field and in the nursery. In contrast, orthodox species exhibited high seed germination (*A. microsperma* and *A. kurzii*). This underscores the importance of maintaining appropriate seed-storage conditions, even for brief durations between seed collection and sowing [19,33].

### 4.3. Seedling Survival and Yield

Seedling yield varied greatly among the species but appeared to have been unaffected by study site location. Seed size affected early post-germination survival. Species with medium-sized seeds (*A. microsperma* and *C. axillaris*) or large ones (*S. pinnata*) achieved high seedling yields, whilst small-seeded species (e.g., *A. fraxinifolius* and *H. dulcis*) attained lower seedling yields. Many previous studies have demonstrated that larger- or intermediate-sized seeds achieve higher seedling-establishment rates than smaller ones [17,36,44], mainly by prolonged provision of stored reserves, which sustain early seedling development and growth [45,46]. This is consistent with the larger-seed-later-commitment mechanism, validated by Kidson and Westoby [47].

Species with high seedling yields tended to have rapid and high seed germination. Rapid germination is highly advantageous, as it reduces the amount of time available for seed predation [14]. It also maximizes the time for root growth before the start of the dry season [48]. This allows roots to access water, deep down in the soil profile, to survive their first dry season and thus greatly reduces first-year mortality [25]. Consequently, to ensure high seedling survival, species characterized by rapid germination with short dormancy periods should be prioritized for direct seeding efforts.

### 4.4. Seedling Growth

*C. axillaris*, *H. dulcis* and *M. azedarach* attained large seedling sizes and high growth rates. They are all pioneer species, which are recommended for forest restoration by the framework tree species method, which involves planting nursery-grown planting stock 30–50 cm tall [25]. In this study, some individual seedlings of *A. microsperma*, *C. axillaris*, *M. azedarach* and *H. dulcis* had grown taller than 30 cm by the end of the study (around 7–8 months after emerging from seeds). Similarly, Tunjai [17] and Waiboonya [19] reported the rapid growth of *M. azedarach* (formerly *M. toosendan*) and *C. axillaris*, with direct-seeded seedlings growing taller than nursery-raised ones due to the better in situ root development. On the other hand, slow-growing species—*C. bakeriana* and *A. fraxinifolius*—produced the smallest seedlings. Even though these two species are categorized as pioneer species [25], they did not perform well in the exposed conditions of the study sites. Therefore, conventional tree planting may be the most suitable way of reintroducing such slow-growing species to degraded areas.

### 4.5. Relative Performance Index (RPI)

This study underscored the importance of appropriate tree-species selection for direct seeding to restore forest ecosystems, emphasizing the need to select species with a combination of attributes, including rapid and high seed germination, which contribute to high seedling yield, and rapid seedling growth. This study was consistent with previous ones [14,17,18,36,48].

*A. microsperma* stood out as the top-performing species (assigned an RPI of 100). The second-best species, *S. pinnata*, achieved an RPI of 50% that of *A. microsperma*. *C. axillaris* and *S. rarak* were considered as acceptable species, with seeds resistant to predation and relatively fast-growing seedlings. On the other hand, species considered unacceptable for direct seeding due to their low RPI were *C. bakeriana*, *G. arborea* and *A. fraxinifolius*. They had low rates of seed germination that resulted in low seedling yields. Their slow-growing, small seedlings could not compete effectively with herbaceous weeds. However, they may potentially be used for direct seeding if seeds are pre-treated to accelerate germination (Table 3). Otherwise, conventional tree planting would be a better option, especially for *G. arborea* and *A. fraxinifolius*, which are considered excellent framework tree species in degraded areas in northern Thailand [25].

### 4.6. Traits to Consider When Selecting Species for Direct Seeding

Our study suggests that species traits can be used to make appropriate species choices for direct seeding, particularly rapid and high germination and high seedling survival and growth, with seeds tolerant of desiccation (orthodox seeds) and of a medium-to-large size. Such criteria should also be considered in combination with site factors. For example, where seed predation is likely to be high, selecting seeds with thick, tough seed coats and sowing them far apart from each other is likely to increase overall success.

Using orthodox seeds for direct seeding offers advantages in terms of both seed availability and pre-sowing storage methods. The seeds of orthodox species are more evenly dispersed throughout the year, with less pronounced seasonality (Figure S3). Such seeds can be dried, stored and sown at the beginning of the rainy season without any loss of viability. Moreover, seed pre-treatments can be applied that accelerate and increase germination (Table 3).

The use of non-orthodox species for direct seeding is more problematic, as indicated by the failure of *A. lacucha*, *G. cowa*, *P. viridis*, *Q. brandisiana*, *S. arboreum* and *S. fruticosum* in this study, despite these species typically exhibiting high seed germination in the nursery [25]. However, the use of such species for direct seeding should not be completely rejected, because including them would greatly enhance the tree species richness of the restored forest ecosystems. Fortunately, most recalcitrant species disperse their seeds at or shortly before the onset of the rainy season (Figure S3)—the optimum time for direct seeding—and germinate rapidly immediately thereafter (including those species listed above). Such

species often fruit prolifically and are easily collected [26]. So, provided they are sown immediately after collection and they are handled with great care between collection and sowing, they may still be used to diversify restoration by direct seeding [19].

**Table 3.** Species-specific recommended practices for direct seeding.

| Species | Collection Month | Sowing Time [a] | Storage Conditions [b] | Seed Pre-Treatments | Light Requirement for Germination | Usefulness [c] |
|---|---|---|---|---|---|---|
| *A. fraxinifolius* | Apr–Jun | RS | RT and RE [4] | Soaking in warm water for 24 h and scarification [2,5] | Full sunlight [2] | Timber |
| *A. microsperma* | Sep–Mar | RS | RT [3] | Without/with scarification [2,4] | Sunlight [4] | Timber, ornamental, dye |
| *A. kurzii* | Jun–Sep | RS | RE [3] | None | Sunlight [4] | Light timber |
| *A. lacucha* | Dec–Jun | IS | - | None [1] | Sunlight [1,4] | Timber, dye, medicinal, flower edible |
| *B. baccata* | Apr–Dec | IS | - | Soak in warm water for 2–3 days [1] | Full sunlight [2,4] | Dye, medicinal, oil, edible fruit |
| *C. bakeriana* | Sep–Jun | RS | RT and RE | Scarification [1] | Sunlight [4] | Timber, ornamental, medicinal |
| *C. axillaris* | Mar–Aug | RS | RE [3] | Soaking in water for 12 h [1] | Sunlight [4] | Edible fruit |
| *D. glandulosa* | May–Oct | RS | RT | Soaking in water for 24 h [1] | Partial shade [1] | Timber, edible fruit |
| *G. cowa* | Sep–Jun | IS | - | No [6] | Shade [4,6] | Medicinal, edible leaf and fruit, varnish |
| *G. arborea* | Mar–Jun | RS | RE [3] | Soaking in water for 12–24 h [1,3,4] | Sunlight [2] | Timber, medicinal |
| *H. dulcis* | Nov–Mar | RS | RE [3] | Soak in water for 1–2 days [1] | Shade [1,5] or 25% sunlight [2] | Medicinal |
| *P. cathia* | Jul–Sep | IS | - | None | - | - |
| *M. azedarach* | Apr–Aug | RS | RE [3] and RT [3,4] | Soak in water for 1–2 days [1] | Sunlight [1,4] | Timber, medicinal, insecticide, edible leaf and flower |
| *M. baillonii* | Aug–Mar | RS | RE [3] | None [1] | Sunlight [2] | Timber |
| *P. emblica* | May–Mar | RS | RT [3] | Scarification [1] | Partial sunlight [1] | Timber, medicinal, edible fruit |
| *P. viridis* | Mar–May | IS | - | None | - | - |
| *P. cerasoides* | Feb–May | RS | RE [3] | None | Sunlight [1,4] | Ornamental, edible fruit, medicinal |
| *Q. brandisiana* | Feb–Jun | IS | - | None | Shade [4] | - |
| *S. rarak* | Jul–Jan | RS | RT and RE | Scarification [1] | Partial sunlight [1,5] or full sunlight [2] | Soap (from fruit) |
| *S. arboreum* | Apr-Jul | IS | - | No [4] | Shade [5] | - |
| *S. pentandrum* | Aug–Oct | RS | - | Scarification [4] | | - |
| *S. pinnata* | Sep–Mar | RS | RT [3] | None | Sunlight [4] | Medicinal, edible flower, fruit, and young leaf |
| *S. fruticosum* | Mar–Aug | IS | - | None [2] | Full sunlight [2] | - |

[1] FORRU [26]; [2] FORRU [25]; [3] Waiboonya [19]; [4] FORRU database; [5] https://plantflowerseeds.com [49] and [6] NPark flora and fauna web [50]. [a] IS = immediately sown at the time of collection, RS = beginning of rainy season; [b] RT = stored at room temperature, RE = stored at 4 °C in a refrigerator [c] Gardner et al. [24].

## 5. Conclusions

For direct seeding in degraded areas in northern Thailand, *A. microsperma*, *S. pinnata* and *C. axillaris* are recommended, due to their low seed predation, high germinability and seedling yield, resulting in high overall performance. For species with low seedling yields, larger quantities of seeds could be collected and sown, and pre-germination techniques could be improved and employed. Moreover, it is important to note that although percent seed removal was low in this study, it is highly variable with site conditions and over time. Therefore, before implementing direct seeding, the potential for seed predation on restoration sites should be considered, and if it is likely to be high, then protective measures should be implemented, such as pelleting and/or burial.

Despite variable success among studies (even in the same location), direct seeding is still expected to be a useful technique for restoring forest ecosystems, although further research is recommended, to improve its efficacy and, in particular, to explore the use of more species and the development of techniques to handle recalcitrant seeds. Furthermore, future climate scenarios that may impact ecosystem restoration efforts should be taken into consideration. Such research may also serve as a foundation for developing aerial seed delivery to upscale restoration.

**Supplementary Materials:** The following supporting information can be downloaded at: https://www.mdpi.com/article/10.3390/f15040674/s1, Figure S1: Two field sites; Ban Mae Khi plot (BMK) and Mon Cham plot (MC); Figure S2: Monthly rainfall and monthly mean temperatures at the Nong Hoi Royal Project station 2019–2021; Figure S3: Seasonal variation in seed dispersal

among recalcitrant vs. orthodox tree species at the community level; Figure S4: Pearson's correlation matrix reveals relationships between field data and seed traits across 23 studied species; Figure S5: Pearson's correlation matrix reveals relationships between field data seed and seedling traits across 13 species with surviving seedlings for nine months; Figure S6: Variables correlation plot generated from principal component analysis (PCA) of field and species-trait data of 13 species with surviving seedlings.

**Author Contributions:** Conceptualization, K.N., S.E. and P.T.; methodology, S.E. and P.T.; investigation, K.N.; formal analysis, K.N. and P.T. resources, W.P.-a.; data curation, K.N.; funding acquisition, K.N., S.E. and P.T.; supervision, W.P.-a., S.E. and P.T.; visualization, K.N.; writing—original draft, K.N., S.E. and P.T.; writing—review and editing, K.N., W.P.-a., S.E. and P.T. All authors have read and agreed to the published version of the manuscript.

**Funding:** This work was supported by a scholarship from the Development and Promotion of Science and Technology Talents Project (DPST), Royal Government of Thailand (ID 562137), funding from the Thailand Research Fund (TRF Grant Number: MRG5980177) and by a small project-support grant from Chiang Mai University's Forest Restoration Research Unit (which covered some of the field work costs). Chiang Mai University's Department of Biology provided some research equipment and laboratory facilities.

**Data Availability Statement:** The data are presented in the Supplementary Materials.

**Acknowledgments:** We thank Nong Hoi Royal Project (Mon Cham and Ban Mae Khi), Ban Nong Hoi, Mae Rim District, Chiang Mai Province, for granting permission to access the study sites and Doi Suthep Pui National Park authority for permission to work in the area. We are grateful to the many volunteers from FORRU, who assisted with field work throughout the study, especially Thonghyod Chiangkantha, Thonglou Pelakul, Jatupoom Meesana and Panya Waiboonya. We also deeply appreciate the suggestions and comments to complete our research from Thanakorn Lattirasuvan and Dia Panitnard Shannon. Finally, we sincerely appreciate the valuable feedback provided by the three reviewers.

**Conflicts of Interest:** The authors declare no conflicts of interest.

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
