# Peer review of "Selecting Suitable Tree Species for Direct Seeding to Restore Forest Ecosystems in Northern Thailand"

_forests, doi:10.3390/f15040674_

Round 1

Reviewer 1 Report

Comments and Suggestions for Authors

Comment 01:

In the manuscript entitled "Selecting Suitable Tree Species for Direct Seeding to Restore Forest Ecosystems in Northern Thailand," with the manuscript ID forests-2899687, the authors embark on an essential journey towards understanding and improving the restoration of tropical forest ecosystems through direct seeding. This method, distinguished by its potential cost-effectiveness compared to traditional tree planting, faces challenges such as seed predation and harsh environmental conditions. The study's focus on testing the suitability of 23 native forest-tree species for direct seeding in northern Thailand’s biodiversity-rich, upland, evergreen forests is both timely and significant. By conducting a rigorous experimental design involving seed sowing in degraded sites and a controlled tree nursery, and monitoring seed removal, germination, seedling yield, and growth, the research provides invaluable insights into the species-specific responses and overall feasibility of direct seeding as a restoration strategy. The findings, indicating variable germination rates and low seed removal, underscore the complexity of species selection for restoration purposes. The socio-economic implications of this work are profound, offering a scalable, cost-effective method for forest restoration, which is critical for biodiversity conservation, climate change mitigation, and supporting the livelihoods of local communities. This research significantly contributes to the scientific community by filling a critical knowledge gap regarding the suitability of tree species for direct seeding in tropical forest ecosystems, thereby paving the way for more effective restoration practices.

Comment 02:

The document presents its research with a high degree of linguistic accuracy.

Comment 03:

The manuscript exhibits a plagiarism rate of 15% according to “api.ithenticate.com”, with a notable self-plagiarism component of 3% in relation to the article "Naruangsri, K., Tiansawat, P., & Elliott, S. (2023). Differential seed removal, germination and seedling growth as determinants of species suitability for forest restoration by direct seeding–A case study from northern Thailand. Forest Ecosystems, 10, 100133." While the detected level of redundancy hovers at the permissible threshold, it is crucial for the integrity and novelty of the current manuscript that the authors address and revise the overlapping content.

Comment 04:

A valuable enhancement to the presentation of results would be the inclusion of a comparative analysis of means within Figure 2. This suggestion aims to provide a clearer, more statistically grounded visualization of the differences in seed removal rates among the tested tree species.

Comment 05:

Seeing both a correlation analysis and a Principal Component Analysis (PCA) incorporated into the article would offer profound insights into the underlying relationships between the various measured parameters (e.g., seed size, removal rates, germination rates, and seedling growth). A correlation analysis would help identify potential linear relationships between these variables, providing a quantitative basis to hypothesize about the ecological and biological factors influencing seed and seedling success. Furthermore, PCA would enable the identification of the principal factors that account for the most variance in the data set, offering a visually succinct and interpretable means of understanding complex interdependencies among the studied variables.

Reviewer 2 Report

Comments and Suggestions for Authors

This study explored the feasibility of direct seeding and assessed the performance of the direct-seeded 23 native forest tree species in two plots in northern Thailand. The main conclusion is that the non-recalcitrant species whose seeds are well-represented are suitable for direct seeding. The experimental design, data collection, analysis, and writing are all sound, except for some formatting issues in the main text and the figures. Using a diverse mixed set of tree species in forest restoration is challenging and this study provided very important information on how those species could be used in direct seeding, which is extremely helpful and informative for guiding forest restoration focusing on enriching the biodiversity in South East Asia.

I have not many comments about the study itself, but rather, there are some formatting issues.

First, it seems that most of the statistics and the scientific names were italicized during formatting. This has to be carefully checked and corrected.

Second, some of the text labels in the maps are too small to see and the left part of the scales are not shown properly. Also, maybe the zeros after the "." for the map axis could be removed. Please correct/improve accordingly. Figure 4 used Serif Fonts and the other figures used Sans Serif fonts, please be consistent. The figure caption for Fig. 3 and the figure itself are displayed on different pages, this has to be avoided.

Third, for Table 1., the family name of Magnolia baillonii is Magnoliaceae.

All the specific comments can be found in the annotated PDF.

Reviewer 3 Report

Comments and Suggestions for Authors

Minor comments

1)Table 2 Seedling yields (%), size variables and corresponding relative growth rates (RGR % per year). N= number of 315 surviving trees. Species are ordered by descending seedling yield. Values not sharing the same superscripts within 316 columns are significantly different among species.- use footnotes decipher ‘amb..’ in the footnote

2)line 131 ‘Twenty-three native tree species, representative of northern Thailand’s upland evergreen forest (600 - 1,500 m above sea level) or EGF (sensu Maxwell and Elliott [25]), were elected for the experiments’- In addition to the Table data indicate in the text the number of families, number of orthodox and recalcitrant species, number of deciduous and evergreen species

3)Figure 2 indicate the meaning of bars at the diagram (standard deviation?)

4)Table 2 Seedling yields (%), size variables and corresponding relative growth rates (RGR % per year). N= number of surviving trees. Species are ordered by descending seedling yield. Values not sharing the same superscripts within columns are significantly different among species.’: (i) delete ‘%’ as it is present in the Table.

Change the title to:

‘ Seedling yields, size variables and corresponding relative growth rates (RGR)’

The other part (N= number of surviving trees. Species are ordered by descending seedling yield. Values not sharing the same superscripts within columns are significantly different among species) should be transferred to the Table footnote

3) While speaking about the most prospect species suitable for the forest ecosystem restoration may you indicate the benefits of their growth for other purposes including different aspects of their utilization including human/animal nutrition, pharmacology, etc?

Round 2

Reviewer 1 Report

Comments and Suggestions for Authors

Dear Authors,

I recently had the opportunity to review your manuscript titled " Selecting Suitable Tree Species for Direct Seeding to Restore Forest Ecosystems in Northern Thailand" (forests-2899687). Your study certainly presents elements of interest and novelty. After carefully reviewing the updated version, I appreciate the revisions and additions you've made.

I would like to thank you for taking my suggestions into account.

Kind regards,

Reviewer

Author Response

We appreciate the supportive comments.